developmental biology

ICR mouse, EDTA, KSOM, micro-droplet, culture system

**Author for correspondence:**
Yun Seok Heo
e-mail: yunsheo@kmu.ac.kr

# Optimized culture systems for the preimplantation ICR mouse embryos with wide range of EDTA concentrations

## Seema Thapa[1], Seung Hee Kang[2] and Yun Seok Heo[1]

[1]Department of Biomedical Engineering, School of Medicine, Keimyung University, 1095 Dalgubeol-daero, Dalseo-gu, Daegu, Republic of Korea
[2]Keimyung University Dongsan Hospital, 1035 Dalgubeol-daero, Dalseo-gu, Daegu, Republic of Korea

YSH, 0000-0002-7038-4025

In this study, *in vitro* preimplantation embryo culture media especially for outbred stock mice (Institute of Cancer Research (ICR)) were optimized with different concentrations of ethylenediaminetetraacetic acid (EDTA). A plot with embryo development rates against EDTA concentrations ranging from 0 to 500 µM showed a unique pattern with two characteristic peaks. Two hundred micromolar was adopted as an optimal concentration of EDTA. The optimized media were also evaluated with two culture systems: conventional large volume culture system (1 ml) and micro-droplet culture system. In the conventional large volume culture system, the blastocyst development rates were compared among three different media (F-10, KSOM and KSOM with the optimized 200 µM EDTA). The rates were 0.4%, 16.7% and 57.6%, respectively. The development rates for the micro-droplet (10 µl) culture system were 73.9%. In conclusion, 200 µM EDTA concentration in 10 µl droplets in the KSOM medium was found as the most suitable culture conditions for ICR mouse embryos, as the blastocyst development rate was higher in the micro-droplet culture system than in the traditional conventional large volume culture system.

## 1. Introduction

The survival rate and development of the embryo *in vitro* depends on the type of culture system, availability and composition of the media in each stage of the embryo development. Therefore, the optimization of *in vitro* culture system is a fundamental requirement in artificial reproductive technology (ART). Over the years, continuous efforts have seen the achievement of milestones in culture systems improvement with notable evolution occurring

in an animal as well as clinical *in vitro* fertilization (IVF) [1]. For decades, different mouse species have been used as an experimental model to develop and modify culture media that later helped to develop culture systems suitable for human embryos [2,3]. However, the most widely employed and well-understood inbred mouse strains in research fail to appropriately account for genetic variation, providing inadequate representation of the genetically diverse human population. Outbred stock mice, a less employed strain, are believed to have a genetic diversity similar to that of humans. The use of outbred mice has greatly increased as they are considered better subjects for biomedical research [4]. However, zygote differentiation in outbred stock mice (ICR mice) gets arrested at 2-cell stage, a phenomenon termed '2-cell block' [5]. This phenomenon of development ceasing at 2-cell stage of embryo development is usually seen during *in vitro* culture of outbred mice oocytes [6]. Sometimes, morula block occurs where the embryos in morula stage cannot reach blastocyst stage, an important developmental stage. This problem rarely occurs during the culture of inbred and F1 strains of mice *in vitro* [7]. Different culture media used for *in vitro* embryo culture contain transitional metal ions like zinc, copper and iron, which produce harmful oxygen radical during chemical reactions [8]. The addition of EDTA in culture media helps to chelate such metal ions and reduces the 2-cell block [9–11]. Presently, there are ongoing efforts to overcome these blocks in outbred mice strains [6,12]. This study reports the optimized culture condition for *in vitro* development of ICR mice embryos using EDTA.

# 2. Material and methods

## 2.1. Animals

ICR stock mice were obtained from Hana Corporation (Republic of Korea). The female mice were 5–7 weeks (21–25 g) whereas the male mice were 7–12 weeks old (30–40 g). Lighting was adjusted to 12 h light/dark cycle and ad libitum food and drinking water were provided.

## 2.2. EDTA preparation

Five hundred millimolar EDTA (Irvine Scientific) was made into different concentrations of EDTA (100, 200, 300, 400 and 500 µM) and mixed in 1 ml culture media. For droplet culture, EDTA was prepared using media itself. All media were filtered through a 0.22 µm filter.

## 2.3. Superovulation and zygote collection

Female mice were superovulated by intraperitoneal injection of 5 IU pregnant mare serum gonadotropin (PMSG; Sigma-Aldrich, G-4877) followed by 5 IU human chorionic gonadotropin (hCG; Life Sciences, 367222–1000IU) 48 h later. They were subsequently mated with male mice of the same strain and examined the following morning for vaginal plugs to confirm that mating had occurred. Female mice were euthanized by cervical dislocation approximately after 16–18 h of post hCG injection. Oviducts were dissected and pooled into HEPES-buffered F-10 media (Gibco, 11550-043) with 10% serum synthetic substitute (SSS; Irvine Scientific, CA 92705. USA). Zygotes were isolated by tearing the ampulla of oviducts, and cumulus cells were removed using hyaluronidase from bovine testes (Sigma-Aldrich). Collected embryos were cultured in respective media.

## 2.4. Embryo culture

Ten per cent of SSS and HEPES were added to F-10 media and were used as handling media for the experiment. The same F-10 media were also used as culture media for embryos. The primary culture medium used in this study was a potassium simplex optimized medium; KSOM medium (Merck, MR-121-D).

### 2.4.1. Experiment 1: conventional large volume culture system (1 ml)

Collected zygotes were distributed randomly in 10–15 groups and were cultured in Falcon IVF one well dish (Irvine Scientific, 353653) containing (i) F-10 media, (ii) 1 ml KSOM culture medium, and (iii) different concentration of EDTA mixed in KSOM culture media.

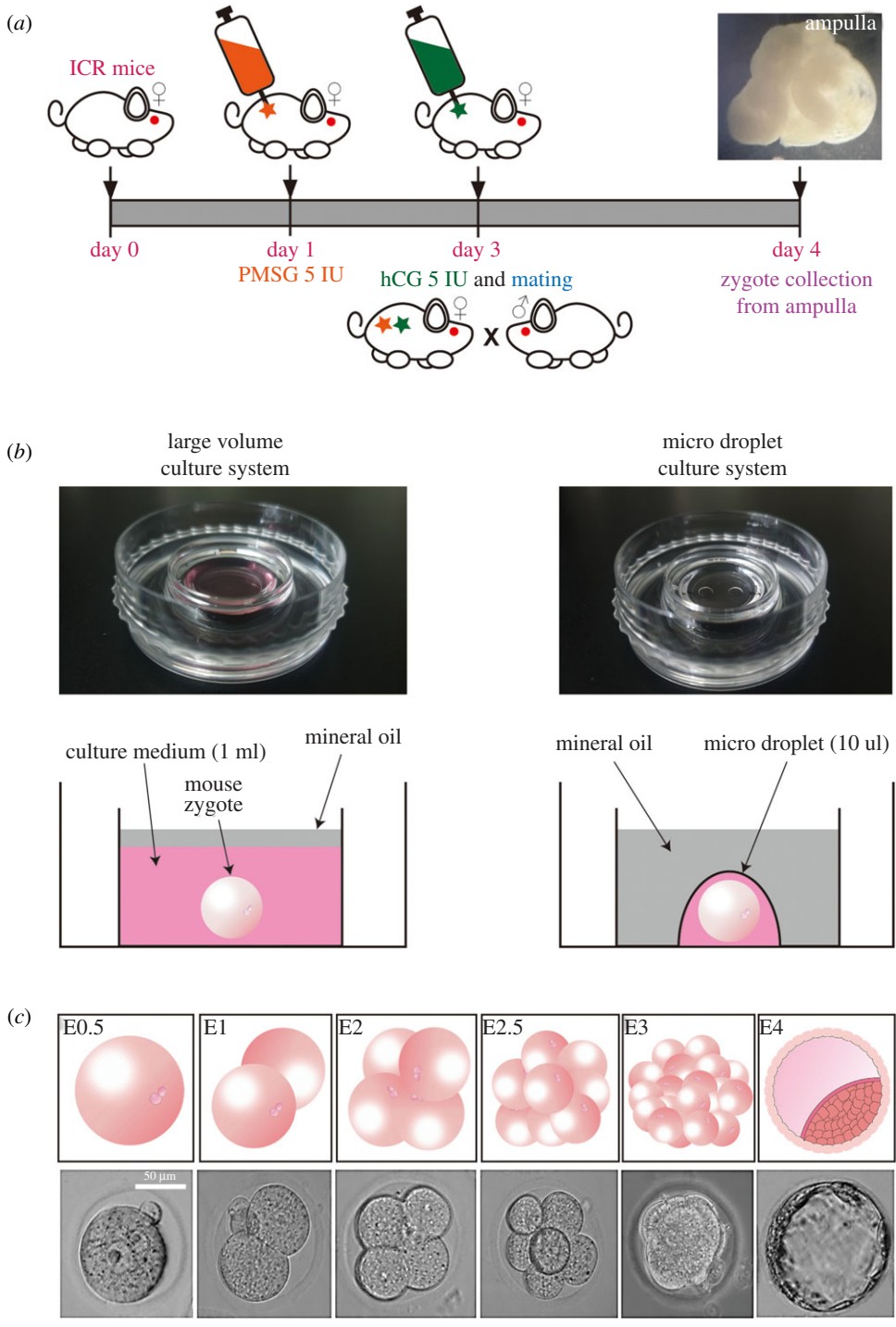

**Figure 1.** Schematic diagram of overall assay procedure. (*a*) Preparation of mice by superovulating with 5 IU of PMSG and hCG, 16–18 h post hCG to collect zygote from ampulla, (*b*) two types of culture systems; large volume culture system and micro-droplet culture system to culture and observe the development of the collected zygotes and (*c*) stages of embryo development observed from Day 0–Day 5.

### 2.4.2. Experiment 2: micro-droplet culture system

The different droplet sizes of media volumes (6.25, 10, 12.5 and 25 µl) were prepared for the culture of embryos. Cultures were performed in both KSOM and EDTA optimized KSOM media. Two droplets of media were made in each Falcon IVF one well dish to culture zygotes in six to nine groups (figure 1).

All culture systems were overlaid with oil for embryo culture (Irvine Scientific, 9305) and were equilibrated overnight before culturing began. Harvested embryos were cultured in the same

incubator with a humidified environment of 5% $CO_2$ at 37°C. Embryo development was assessed every day from 1-cell stage to the formation of blastocyst.

## 2.5. Statistical analysis

Pictures used in this paper were taken using Zeiss Zen 2.5 microscope imaging software. Graphs and percentage of blastocyst development were prepared using Excel 2016 and Origin pro 2019. Percentage data were arcsine transformed before analysis but are presented as non-transformed mean ± SEM unless otherwise mentioned. Data were analysed using ANOVA followed by Tukey honestly significant difference test. A value of $p \leq 0.05$ was considered statistically significant.

# 3. Results

## 3.1. Experiment 1: conventional large volume culture system (1 ml)

### 3.1.1. Comparison between different concentrations of EDTA

A plot of EDTA concentrations against the embryonic development rate showed two distinctive peaks, where the development rate increased steadily from 0–200 µM concentration but decreased drastically at a concentration of 300 µM. However, a slight increase in development rate was seen with increasing EDTA concentration (400–500 µM) at a lower rate than the development rate of 200 µM EDTA. The 57.6% of 92 zygotes were developed to blastocysts in KSOM media with 200 µM of EDTA, whereas, among 508 zygotes, only 16.7% were developed to blastocysts in KSOM media alone figure 2a. Even though most EDTA concentrations supported proper development and survival of embryos up to the morula stage, most of them showed 'morula block' leading to a low blastocyst development rate. Two hundred micromolar proved to be the best EDTA concentration for media optimization.

### 3.1.2. Comparison between culture media

Among three media used to culture embryos in conventional large volume culture system (F-10, KSOM, KSOM + 200 µM EDTA), KSOM with 200 µM EDTA showed the highest blastocyst development. Among 92 zygotes, 57.6% developed into a blastocyst in this media. Whereas F-10 and KSOM media alone showed '2-cell block' and 'morula block', respectively. Only 0.4% of the 501 zygotes in F-10 media developed to blastocysts. Similarly, in KSOM media, only 16.7% of 508 zygotes developed to blastocysts which were 40.9% less than the blastocysts development rate in KSOM media with 200 µM EDTA ($p < 0.01$).

## 3.2. Experiment 2: micro-droplet culture system

### 3.2.1. Micro-droplet culture system optimization

Among the different micro-droplet culture volumes (6.25, 10, 12.5 and 25 µl) used for embryo culture (six embryos/droplet-triplicate readings), the largest culture volume (25 µl) of KSOM media showed the lowest blastocyst development rate. Despite the fact that a quarter (25.2%) of embryos survived up to the morula stage, only 8.3% survived to the blastocyst stage. Similarly, the lowest culture volume (6.25 µl) also weakly supported the development of embryos to blastocysts (18.3%). Ten microliters and 12.5 µl showed the highest blastocyst development rates (25.0%) as shown in figure 2b. Therefore, we chose a 10 µl culture volume for EDTA optimization for further studies. A total number of 18 embryos were used in each case.

Like the conventional large volume culture system, different concentrations of EDTA (0–500 µM) were added to KSOM media to culture mouse embryo in 10 µl micro-droplet and a similar growth pattern was seen in stages preceding the morula stage. Most of the 10 µl micro-droplets made of the KSOM with different EDTA concentrations showed decreased 'morula block' and increased blastocyst development rates as compared with the 10 µl micro-droplets of KSOM media alone. Among these EDTA concentrations, 73.9% of 45 zygotes developed blastocysts in culture media with 200 µM EDTA making it optimum culture concentration. KSOM media alone developed 29.6% of 40 zygotes to blastocysts which were 44.3% ($p < 0.01$) less than KSOM + 200 µM EDTA medium as shown in figure 2c.

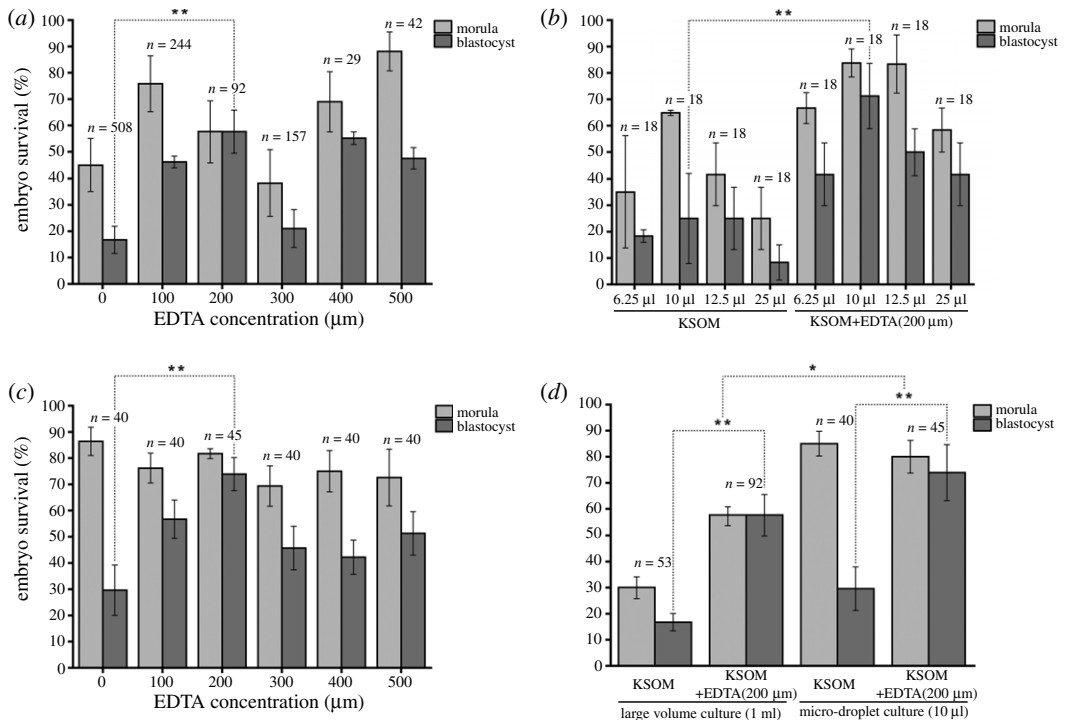

**Figure 2.** Charts of embryo survival rates (morula and blastocyst) (*a*) Morula and blastocyst development rate of embryos cultured in KSOM (0) and different concentrations of EDTA (0–500 µM) in large volume culture system (1 ml), (*b*) development rate of morula and blastocyst in different droplet media volume in KSOM alone and EDTA optimized KSOM media, (*c*) development rate of embryos in KSOM and EDTA optimized KSOM media in 10 µl micro-droplet culture system and (*d*) comparison of morula block and blastocyst development rate in two culture systems with KSOM media alone and KSOM media with 200 µM EDTA. **Significantly different to medium KSOM (0); $p < 0.01$; *Significantly different in between two culture systems; $p < 0.05$.

In figure 2*b*, after the optimization of EDTA concentration, the effect of different droplet volumes (6.25, 10, 12.5 and 25 µl) was further evaluated with KSOM + EDTA (200 µM) simultaneously ($n = 18$ each). In four different volumes of culture media, the pattern of growth was similar to that of KSOM media alone, however, with a higher rate of blastocyst development. Blastocyst development rates in the lowest and highest culture volume (6.25, 25 µl) in KSOM + EDTA (200 µM) media were 41.6%. The highest 'morula block' was seen in the lowest culture volume. The difference in the blastocyst development rate of the two media (KSOM with 200 µM EDTA versus KSOM media alone) in 10 µl micro-droplet was 48% (73.9% versus 25.9%), making it a suitable volume size for embryo development ($p < 0.01$).

## 3.3. Comparison between culture systems

As shown in figure 2*d*, the rate of blastocyst developments in the micro-droplet culture system was seen to be higher than in the conventional large volume culture system after 96 h of culture. The blastocyst development rate in EDTA optimized media was improved by 16.3% (73.9% versus 57.6%) in 10 µl micro-droplet culture system ($p < 0.05$). However, the optimum concentration of EDTA was the same (200 µM) in both culture systems. Blastocyst development rate in KSOM media alone was also higher in the micro-droplet culture system compared with conventional large volume culture system by 12.9% (29.6% versus 16.7%) ($p < 0.05$).

## 4. Discussion

Since IVF technique was established, numerous media formulations, as well as culture systems for *in vitro* embryo preimplantation have been developed. The use of inbred mice is well known; however, the use of outbred mice such as ICR mice is slowly making its way to the different fields of research [13]. Nonetheless, the studies related to *in vitro* culture of such mouse embryos are few because the

development block occurs when one-cell embryos are cultured in conventional media. If the culture system for ICR mouse is well developed, ICR mice can be used as a good model for ART as they are considered to have genetic variations similar to humans. Besides, ICR mice are docile, well-developed and cost effective. In the present study, we experimented with F-10 media, KSOM and KSOM with different concentrations of EDTA. EDTA in culture media acts as a chelating agent and enhances the development and differentiation of zygotes to blastocyst. The types of commercially available culture media have a different range of EDTA concentration that works for the embryo of different mice strain and species. Our experiment aimed to find out the optimum range of EDTA in KSOM media using ICR mice embryos. We examined EDTA concentration ranging from 0 to 500 µM in both conventional large volume and micro-droplet culture systems. In both culture systems, two distinctive peaks were observed when embryonic development rates were plotted from 0 to 500 µM concentrations of EDTA. The second peak at 500 µM EDTA concentration showed a significant embryo development and survival rate up to the morula stage but a reduced blastocyst development rate. At 200 µM EDTA concentration, the first peak showed a higher blastocyst development rate than the second peak. However, the reason for the varying development rates was not studied.

A significant increase in cleavage rate was observed in EDTA optimized media. Whereas the embryos cultured in F-10 and KSOM media alone showed the '2-cell' and 'morula block', respectively. In conventional large volume culture system F-10 media, out of 501 zygotes, only 142 (28.3%) of them made it to 2-cell stage. The blastocyst development rate was also low at 2 (0.4%). A further experiment was not carried out due to the extremely low blastocyst development. Likewise, in KSOM alone media, out of 229 morula stage embryo, only 85 (37.1%) of them developed to the blastocyst stage. Addition of EDTA in culture media subsequently increased the blastocyst development rate in both culture systems. Blastocyst development rate in EDTA optimized media was increased by 40.9% and by 48% in large and micro-droplet culture system, respectively, ($p < 0.01$) than in KSOM media alone. As reported by previous studies, EDTA reduces the developmental block through the chelation of metal ions in embryo culture [14] with its capacity to neutralize metal toxicity caused by heavy metals [15]. A decrease in the glycolytic activity of 2-cell stage mice embryo in presence of EDTA also explains its beneficial effect [16]. An increase in glycolytic activity reduces embryo development, which is seen to have increased in the absence of EDTA in culture media [17]. Our present study also proved these beneficial effects of EDTA. Our study also found the 10 µl droplet media with 200 µM EDTA in KSOM to be a more efficient culture condition than higher media volume (1 ml, 25 µl) and minimum media volume (6.25 µl) for ICR mice embryos. Our findings were in agreement with a previous study that reported that the rate of embryo development improves when the embryos are culture in reduced culture volume or increased number of embryos [18,19]. It has been reported that the production of growth factors like autocrine and paracrine mutually benefit embryos in group culture [20]. However, a minimal volume of culture media can also be detrimental to the embryos due to toxic metabolite production.

Our study also supports the fact that the higher concentration of EDTA in culture media reduces the development of embryos as suggested by Matsukawa *et al*. [21]. In this article, the EDTA concentration of 10 µM was optimized followed by 100 µM in ICR mice embryos cultured also in KSOM as basal media. However, the study was not conducted with more than 100 µM EDTA concentration except for 1000 µM (10, 100 and 1000 µM). The number of embryos per 50 µl droplet is also unknown. By contrast, our study tried to be more specific about the culture volume and embryo number along with a wide range of EDTA concentrations. However, some of the data in our present study may vary with the referenced previous studies; this could be due to the difference in embryo number, culture media, media volume and concentrations of EDTA. This study can contribute to set the ICR mouse as a good model species for IVF study.

# 5. Conclusion

Commercially purchased KSOM medium was optimized, where 10 µl of KSOM with 200 µM of EDTA was found to have the most suitable conditions to culture ICR mouse embryos. This study's findings are useful in embryology; they provide insight into the improvement of human embryo culture technique, especially, for research studies looking for genetically diverse options. They are also useful to other fields that would benefit from genetic diversity such as toxicology and pharmacology.

Ethics. This study was approved by the Institutional Animal Care and the Use Committee of the School of Medicine, Keimyung University (Mice IRB no. KM-2017-44R1) and was performed following the relevant guidelines and regulations.

Data accessibility. Data are available from the Dryad Digital Repository: https://doi.org/10.5061/dryad.sqv9s4n2g [22].

Authors' contributions. Y.S.H. designed the project and supervised the research; S.T. and S.H.K. conducted the experiments and analysed the results; S.T. and Y.S.H. wrote the manuscript; all authors contributed to discussions about the research and reviewed the manuscript.

Competing interests. We declare we have no competing interests.

Funding. This research was majorly supported by the National Research Foundation (NRF) of Korea, funded by the Korean Government (MSIP) (no. 2014R1A5A2010008), (MSIT) (no. 2017R1A2B1011004) (no. 2020R1F1A1066348). It was also partly supported by a grant from the Korea Health Technology R&D Project through the Korea Health Industry Development Institute (KHIDI), funded by the Ministry of Health and Welfare, the Republic of Korea (no. HI14C1324).

Acknowledgements. We thank Dr Polly Campbell and two anonymous reviewers for their valuable comments which helped us improve this manuscript. Further, we sincerely thank our colleague, Ryong Sung, for his contribution in the manuscript discussion.

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
