## [Peer Review File · Royal Society Open Science]

Review History

RSOS-201752.R0 (Original submission)

Review form: Reviewer 1

Is the manuscript scientifically sound in its present form?

Yes

Are the interpretations and conclusions justified by the results?

Yes

Is the language acceptable?

No

Do you have any ethical concerns with this paper?

No

Have you any concerns about statistical analyses in this paper?

Yes

Recommendation?

Major revision is needed (please make suggestions in comments)

Comments to the Author(s)

Thapa et al. present a short report on their embryo culture media optimization. I believe that these reports are very valuable to the specialist community and well worth publishing. The approach seems appropriate and straight forward, however, I have a few comments/concerns.

Major comments:

There have been studies that looked at different EDTA concentration in embryo culture. Notably Matsukawa et al. 2002 in ICR embryos. Of course, this does not impede the publication of a similar study, but the authors should mention this study and compare their results with their own.

Matsukawa, T., Ikeda, S., Imai, H. and Yamada, M., 2002. Alleviation of the two-cell block of ICR mouse embryos by polyaminocarboxylate metal chelators. *Reproduction* (Cambridge, England), 124(1), pp.65-71.

Fig. 2a: The sample sizes vary quite a bit. Were the same number of isolated zygotes used for each condition? If so, does the n reported here refer to the number that developed to 2-cell stage? Could the medium condition cause these differences?

Introduction: Few more sentences introducing the mouse strain and its uses would be very helpful to readers. Also, more information about the 2-cell block and the morula-block and how it can be overcome with EDTA would be useful in the introduction.

Section 3.1.2., line 27-28 and Section 3.2.1., line 58-59: The authors should provide a p-values for these comparisons.

Minor comments:

Title: I do not think the quotes are necessary.

Line 48: which mouse strain/strains?

The text has grammatical mistakes throughout the manuscript that should be corrected.

Review form: Reviewer 2

Is the manuscript scientifically sound in its present form?

Yes

Are the interpretations and conclusions justified by the results?

Yes

Is the language acceptable?

No

Do you have any ethical concerns with this paper?

No

Have you any concerns about statistical analyses in this paper?

Yes

Recommendation?

Major revision is needed (please make suggestions in comments)

Comments to the Author(s)

Heo and co-workers optimized media condition of EDTA concentration to increase embryo development rate for preimplantation. The optimized media were tested in two culture systems: conventional large volume and micro-droplet systems. Experimental results can be of interest. However, more discussion about the mechanism and reason for the experimental data can be helpful. Below are specific comments and critiques on the manuscript.

- The manuscript needs to correct grammatical errors and typos. E.g. 'preimplatation' 'with wide range of ...'
- It should be better to describe a full name before using its abbreviation. e.g. IVF
- In line 42-43, page 3, it seems there is a grammatical error.
- In line 23, page 4, they used tested for three types of media. However, a result for F-10 was not shown in the plot in Fig.2. The explanation why the result for F-10 was not shown was not clear.
- In line 59-62, page 4, the sentence is difficult to read.
- In line 5, page 5, 'improved by 16.0%'. The difference between two groups is not 16.0%. It can be calculated more precisely.
- Statistical differences between groups can be included in plots in Fig.2.
- In line 17-19, page 5, the sentence is difficult to read.
- In line 19-21, page 5, the sentence is difficult to read.
- References in bibliography should have the same format of journal name.

Decision letter (RSOS-201752.R0)

Dear Dr Heo

On behalf of the Editors, we are pleased to inform you that your Manuscript RSOS-201752 "Optimized culture systems for the Preimplatation ICR mouse embryos with "wide" range of EDTA concentrations" has been accepted for publication in Royal Society Open Science subject to minor revision in accordance with the referees' reports. Please find the referees' comments along with any feedback from the Editors below my signature.

Please submit your revised manuscript and required files (see below) no later than 7 days from today's (ie 02-Mar-2021) date. Note: the ScholarOne system will 'lock' if submission of the revision is attempted 7 or more days after the deadline. If you do not think you will be able to meet this deadline please contact the editorial office immediately.

on behalf of Dr Polly Campbell (Associate Editor) and Kevin Padian (Subject Editor)
openscience@royalsociety.org

Associate Editor Comments to Author (Dr Polly Campbell):

Comments to the Author:

First, I want to apologize to the authors for the long delay in getting a decision on their manuscript. Both reviewers see your study as a useful contribution and both make valuable suggestions for improvement. I encourage the authors to take these critiques into account in their revision and have added some additional suggestions and comments below.

Introduction

Add citations to support the series of statements in the sentences beginning "Over the years..." and "For decades, ..."

In the sentence that follows ("However, the most widely...") indicate what mouse strain you are referring to. Or is this referring to inbred lab mice in general? Please clarify.

"However zygote differentiation in ICR mice gets arrested at 2 cell stage..." Please clarify that you're referring to what happens in vitro.

Section 2.1

Lighting was adjusted

Section 2.3

What was the site of the injection? i.e. was it subcutaneous or other?

Section 2.4.2

First sentence is hard to understand. Please try to revise for clarity.

Results

Some but not all sample sizes are reported in the text. Please be consistent.

Section 3.1.2

This whole section is very hard to follow. Suggest starting with a sentence that reports the result for the best media combo, including % of zygotes that made it to blastocyst stage

Section 3.2.1

Avoid the word “significant” in the absence of statistics. Could just say, “...despite the fact that a quarter (25.2%) of embryos survived....”

Please fix grammar in sentence starting, “The lowest culture volume..”

Section 3.3

I think it should be 73.9% vs. 57.6% and 29.6% vs. 16.7%.

Please report statistical test in addition to the p-value.

Discussion

Sentence starting “However, the concentration...” is hard to understand. Are the authors referring to variation in EDTA concentration in commercial media or something else? Also not clear what “EDTA concentration efficient range” means. Please clarify. Suggest replacing “huge” with large or wide.

Fig. 2

Rate refers to change in a variable measured over time. What is shown here is the percent of embryos that reach the target stages. Suggest changing Y axis labels to something like % Embryo Survival or % Surviving Embryos and adjusting caption accordingly.

Reviewer comments to Author:

Reviewer: 1

Comments to the Author(s)

Thapa et al. present a short report on their embryo culture media optimization. I believe that these reports are very valuable to the specialist community and well worth publishing. The approach seems appropriate and straight forward, however, I have a few comments/concerns.

Major comments:

There have been studies that looked at different EDTA concentration in embryo culture. Notably Matsukawa et al. 2002 in ICR embryos. Of course, this does not impede the publication of a similar study, but the authors should mention this study and compare their results with their own.

Matsukawa, T., Ikeda, S., Imai, H. and Yamada, M., 2002. Alleviation of the two-cell block of ICR mouse embryos by polyaminocarboxylate metal chelators. *Reproduction* (Cambridge, England), 124(1), pp.65-71.

Fig. 2a: The sample sizes vary quite a bit. Were the same number of isolated zygotes used for each condition? If so, does the n reported here refer to the number that developed to 2-cell stage? Could the medium condition cause these differences?

Introduction: Few more sentences introducing the mouse strain and its uses would be very helpful to readers. Also, more information about the 2-cell block and the morula-block and how it can be overcome with EDTA would be useful in the introduction.

Section 3.1.2., line 27-28 and Section 3.2.1., line 58-59: The authors should provide a p-values for these comparisons.

Minor comments:

Title: I do not think the quotes are necessary.

Line 48: which mouse strain/strains?

The text has grammatical mistakes throughout the manuscript that should be corrected.

Reviewer: 2

Comments to the Author(s)

Heo and co-workers optimized media condition of EDTA concentration to increase embryo development rate for preimplantation. The optimized media were tested in two culture systems: conventional large volume and micro-droplet systems. Experimental results can be of interest. However, more discussion about the mechanism and reason for the experimental data can be helpful. Below are specific comments and critiques on the manuscript.

- The manuscript needs to correct grammatical errors and typos. E.g. 'preimplatation' 'with wide range of ...'
- It should be better to describe a full name before using its abbreviation. e.g. IVF
- In line 42-43, page 3, it seems there is a grammatical error.
- In line 23, page 4, they used tested for three types of media. However, a result for F-10 was not shown in the plot in Fig.2. The explanation why the result for F-10 was not shown was not clear.
- In line 59-62, page 4, the sentence is difficult to read.
- In line 5, page 5, 'improved by 16.0%'. The difference between two groups is not 16.0%. It can be calculated more precisely.
- Statistical differences between groups can be included in plots in Fig.2.
- In line 17-19, page 5, the sentence is difficult to read.
- In line 19-21, page 5, the sentence is difficult to read.
- References in bibliography should have the same format of journal name.

===PREPARING YOUR MANUSCRIPT===

- one version identifying all the changes that have been made (for instance, in coloured highlight, in bold text, or tracked changes);
- a 'clean' version of the new manuscript that incorporates the changes made, but does not highlight them. This version will be used for typesetting.

If you have been asked to revise the written English in your submission as a condition of publication, you must do so, and you are expected to provide evidence that you have received language editing support. The journal would prefer that you use a professional language editing service and provide a certificate of editing, but a signed letter from a colleague who is a native

speaker of English is acceptable. Note the journal has arranged a number of discounts for authors using professional language editing services (<https://royalsociety.org/journals/authors/benefits/language-editing/>).

===PREPARING YOUR REVISION IN SCHOLARONE===

-- If you have uploaded ESM files, please ensure you follow the guidance at <https://royalsociety.org/journals/authors/author-guidelines/#supplementary-material> to include a suitable title and informative caption. An example of appropriate titling and captioning

may be found at https://figshare.com/articles/Table_S2_from_Is_there_a_trade-off_between_peak_performance_and_performance_breadth_across_temperatures_for_aerobic_sc_ope_in_teleost_fishes_/3843624.

Author's Response to Decision Letter for (RSOS-201752.R0)

See Appendix A.

Decision letter (RSOS-201752.R1)

Dear Dr Heo

On behalf of the Editors, we are pleased to inform you that your Manuscript RSOS-201752.R1 "Optimized culture systems for the preimplantation ICR mouse embryos with wide range of EDTA concentrations" has been accepted for publication in Royal Society Open Science subject to minor revision in accordance with the referees' reports. Please find the referees' comments along with any feedback from the Editors below my signature.

Please submit your revised manuscript and required files (see below) no later than 7 days from today's (ie 15-Mar-2021) date. Note: the ScholarOne system will 'lock' if submission of the revision is attempted 7 or more days after the deadline. If you do not think you will be able to meet this deadline please contact the editorial office immediately.

Kind regards,
Royal Society Open Science Editorial Office

on behalf of Dr Polly Campbell (Associate Editor) and Kevin Padian (Subject Editor)
 openscience@royalsociety.org

Associate Editor Comments to Author (Dr Polly Campbell):

Associate Editor

Comments to the Author:

My comments and those of the reviewers have been adequately addressed. Please make the following minor changes prior to final submission:

Introduction

L36 fail not fails

L42-44 Suggest deleting sentence beginning "Due to lack". It mostly repeats what's already been said.

L1 add "the" before addition

Section 2.1

L11 Lighting not lightening

Section 3.2.1

Thank you for adding sample sizes to section 3.1.2. Please do the same here. NB that percentages are not sample sizes.

L42 Change to, "...only 8.3% survived to the blastocyst..."

===PREPARING YOUR MANUSCRIPT===

===PREPARING YOUR REVISION IN SCHOLARONE===

Author's Response to Decision Letter for (RSOS-201752.R1)

See Appendix B.

Decision letter (RSOS-201752.R2)

Dear Dr Heo,

I am pleased to inform you that your manuscript entitled "Optimized culture systems for the preimplantation ICR mouse embryos with wide range of EDTA concentrations" is now accepted for publication in Royal Society Open Science.

If you have not already done so, please remember to make any data sets or code libraries 'live' prior to publication, and update any links as needed when you receive a proof to check - for instance, from a private 'for review' Dryad URL to the publicly accessible 'for publication' DOI. It is good practice to also add data sets, code and other digital materials to your reference list.

Best regards,

on behalf of Dr Polly Campbell (Associate Editor) and Kevin Padian (Subject Editor)
openscience@royalsociety.org

Appendix A

Response to Editor and Referees comments

We would like to thank all the reviewers for their valuable input in our manuscript. Here are the responses to the comments made.

Associate Editor Comments to Author (Dr. Polly Campbell):

Comments to the Author:

First, I want to apologize to the authors for the long delay in getting a decision on their manuscript. Both reviewers see your study as a useful contribution and both make valuable suggestions for improvement. I encourage the authors to take these critiques into account in their revision and have added some additional suggestions and comments below.

Introduction

Add citations to support the series of statements in the sentences beginning "Over the years..." and "For decades, ..."

Answer: The citations are added in the sentences beginning with "over the years..." and "for decades..." in introduction section, line 32 and line 34.

In the sentence that follows ("However, the most widely...") indicate what mouse strain you are referring to. Or is this referring to inbred lab mice in general? Please clarify.

Answer: The sentence was meant to indicate the inbred mice in general as widely employed and well understood strain. The sentence clarifying the mouse strain is added in introduction section line 34.

"However zygote differentiation in ICR mice gets arrested at 2 cell stage..." Please clarify that you're referring to what happens in vitro.

Answer: Unlike inbred mouse embryos, outbred mouse embryos cannot differentiate further to blastocyst stage from 2 cell stage due to the lack of *in vivo* like condition while cultured *in vitro*. Few more sentences are added in the introduction section line 39-45 (Page 1) and line 1-2 (Page 2) to clarify the confusion.

Section 2.1

Lighting was adjusted

Answer: Correction is made to lighting was adjusted from "lighting were adjusted" in section 2.1., line 11.

Section 2.3

What was the site of the injection? i.e. was it subcutaneous or other?

Answer: The site of the injection was intraperitoneal. The site of injection is added in section 2.3., line 21.

Section 2.4.2

First sentence is hard to understand. Please try to revise for clarity.

Answer: The sentence is revised for clarity as "The different droplet sizes of media volumes (6.25, 10, 12.5, and 25 μ l) were prepared for the culture of embryos. Cultures were performed in both KSOM and EDTA optimized KSOM media" in section 2.4.2., line 45-46.

Results

Some but not all sample sizes are reported in the text. Please be consistent.

Answer: Sample sizes are reported throughout the result section. Especially from section 3.1. to 3.2.

Section 3.1.2

This whole section is very hard to follow. Suggest starting with a sentence that reports the result for the best media combo, including % of zygotes that made it to blastocyst stage

Answer: The whole section 3.1.2. is restructured as "Among three media used to culture embryos in conventional large volume culture system (F-10, KSOM, KSOM + 200 μ M EDTA), KSOM with 200 μ M EDTA showed highest blastocyst development. Among 92 zygotes, 57.6% developed into blastocyst in this media. Whereas, F-10 and KSOM media alone showed "2- cell block" and "morula block" respectively. Only 0.4% of the 501 zygotes in F-10 media developed to blastocysts. Similarly, in KSOM media, only 16.7% of 508 zygotes developed to blastocysts which were 40.9% less than the blastocysts development rate in KSOM media with 200 μ M EDTA ($P < 0.01$)" for more clarity.

Section 3.2.1

Avoid the word "significant" in the absence of statistics. Could just say, "...despite the fact that a quarter (25.2%) of embryos survived..."

Answer: The sentences are restructured as "Despite the fact that a quarter (25.2%) of embryos survived up to morula stage, only 8.3% of it made to the blastocyst stage" in section 3.2.1., line 41-42.

Please fix grammar in sentence starting, "The lowest culture volume.."

Answer: The sentence is rewritten as " the lowest culture volume (6.25µl) also weakly supported the development of embryos to blastocysts (18.3%)" in section 3.2.1., line 42-43.

Section 3.3

I think it should be 73.9% vs. 57.6% and 29.6% vs. 16.7%.

Answer: The correction is made from (57.6% vs.73.9%) and (16.7% vs. 29.6%) to (73.9% vs. 57.6%) and (29.6% vs. 16.7%) in section 3.3., line 14 and 18.

Please report statistical test in addition to the p-value.

Answer: For the statistical test percentage data were arcsine-transformed. Data were analyzed using ANOVA followed by Tukey honestly significant difference test. The statistical test is added in the statistical analysis section 2.5., line 3-5 (Page 3).

Discussion

Sentence starting "However, the concentration..." is hard to understand. Are the authors referring to variation in EDTA concentration in commercial media or something else? Also not clear what "EDTA concentration efficient range" means. Please clarify. Suggest replacing "huge" with large or wide.

Answer: The sentence is shortened to "The types of commercially available culture media have a different range of EDTA concentration that works for embryo of different mice strain and species" in discussion section, line 33-34 for more clarity.

Fig. 2

Rate refers to change in a variable measured over time. What is shown here is the percent of embryos that reach the target stages. Suggest changing Y axis labels to something like % Embryo Survival or % Surviving Embryos and adjusting caption accordingly.

Answer: The statement is true indeed. The y-axis labels are changed to Embryo survival (%) and caption is adjusted accordingly in figure 2.

Reviewer comments to Author:

Reviewer: 1

Comments to the Author(s)

Thapa et al. present a short report on their embryo culture media optimization. I believe that these reports are very valuable to the specialist community and well worth publishing. The approach seems appropriate and straight forward, however, I have a few comments/concerns.

Major comments:

There have been studies that looked at different EDTA concentration in embryo culture. Notably Matsukawa et al. 2002 in ICR embryos. Of course, this does not impede the publication of a similar study, but the authors should mention this study and compare their results with their own.

Matsukawa, T., Ikeda, S., Imai, H. and Yamada, M., 2002. Alleviation of the two-cell block of ICR mouse embryos by polyaminocarboxylate metal chelators. Reproduction (Cambridge, England), 124(1), pp.65-71.

Answer: The study done by the Matsukawa et.al. is similar to the current study indeed. The comparison wherever possible is made as suggested in discussion section line 6-12 (Page 5).

Fig. 2a: The sample sizes vary quite a bit. Were the same number of isolated zygotes used for each condition? If so, does the n reported here refer to the number that developed to 2-cell stage? Could the medium condition cause these differences?

Answer: The n number reported here is total number of isolated zygotes used for the experiment. The sample size was increased in some cases to confirm the results. The increase in sample size also didn't help the result much. The development of zygotes to 2-cell and to blastocyst stage was definitely affected by the medium condition, presence and absence of EDTA in culture media. Few sentences regarding this is added in discussion section page 4, line 45-51.

Introduction: Few more sentences introducing the mouse strain and its uses would be very helpful to readers. Also, more information about the 2-cell block and the morula-block and how it can be overcome with EDTA would be useful in the introduction.

Answer: Few sentences are added as suggested on both mouse strain, 2-cell block and the function of EDTA in overcoming the developmental blocks in introduction section page 1, line 39-45 and page 2, line 1-2.

Section 3.1.2., line 27-28 and Section 3.2.1., line 58-59: The authors should provide a p-values for these comparisons.

Answer: The *P*- values are provided in section 3.1.2., line 33 and in section 3.2.1., line 54 (Page 3) and line 8 (page 4).

Minor comments:

Title: I do not think the quotes are necessary.

Answer: The quotes are removed from the title.

Line 48: which mouse strain/strains?

The ICR mouse strain is added in the section 2.1., line 10.

The text has grammatical mistakes throughout the manuscript that should be corrected.

Answer: The manuscript is revised again to minimize the grammatical mistakes.

Reviewer: 2

Comments to the Author(s)

Heo and co-workers optimized media condition of EDTA concentration to increase embryo development rate for preimplantation. The optimized media were tested in two culture systems: conventional large volume and micro-droplet systems. Experimental results can be of interest. However, more discussion about the mechanism and reason for the experimental data can be helpful. Below are specific comments and critiques on the manuscript.

Answer: Some more points in the discussion section are added as suggested from line 24-29 (Page 4).

- The manuscript needs to correct grammatical errors and typos. E.g. 'preimplantation' 'with wide range of ...'

Answer: The manuscript is revised again to minimize the grammatical errors and typos.

- It should be better to describe a full name before using its abbreviation. e.g. IVF

Answer: The corrections are made accordingly in introduction section, line 32.

- In line 42-43, page 3, it seems there is a grammatical error.

Answer: The sentence is restructured to make it more clear in section 2.4.2., line 45-46.

- In line 23, page 4, they used tested for three types of media. However, a result for F-10 was not shown in the plot in Fig.2. The explanation why the result for F-10 was not shown was not clear.

Answer: In case of F-10 media, very few embryos (0.4%) made up to blastocyst stage despite of 501 embryos experimented. The blastocyst development was quite low so, it is not mentioned in the figure. Few sentences to clear the confusion is added in the discussion section, line 45-48 (Page 4).

- In line 59-62, page 4, the sentence is difficult to read.

Answer: The line is restructured for better understanding as "blastocyst development rates in the lowest and highest culture volume (6.25, 25 μ l) in KSOM + EDTA (200 μ M) media were 41.6%. The highest "morula block" was seen in the lowest culture volume. The difference in blastocyst development rate of the two media (KSOM media alone vs. KSOM with 200 μ M EDTA) in 10 μ l micro-droplet was 48% (73.9% vs. 25.9%), making it a suitable volume size for embryo development ($P < 0.01$)" in page 4 line 4-8.

- In line 5, page 5, 'improved by 16.0%'. The difference between two groups is not 16.0%. It can be calculated more precisely.

Answer: The difference is calculated precisely and is changed to 16.3% in section 3.3., line 14 (Page 4).

- Statistical differences between groups can be included in plots in Fig.2.

Answer: The statistical differences between groups are included in plots in Figure as suggested.

- In line 17-19, page 5, the sentence is difficult to read.

Answer: The sentence is corrected to make it easier to read in discussion section, line 30-33 (Page 4).

- In line 19-21, page 5, the sentence is difficult to read.

Answer: The sentence is rewritten as "The types of commercially available culture media have a different range of EDTA concentration that works for embryo of different mice strain and species" in discussion section, line 33-34 (Page 4).

- References in bibliography should have the same format of journal name.

Answer: The journal names are corrected to match the same format in reference section.

Appendix B

Responses to Associate Editor Comments

We would like to thank Dr. Polly Campbell again for these valuable comments to improve our manuscript. Here are the responses to the corrections we have made as suggested.

Associate Editor Comments to Author (Dr. Polly Campbell): Associate Editor

Comments to the Author:

My comments and those of the reviewers have been adequately addressed. Please make the following minor changes prior to final submission:

Introduction

L36 fail not fails

Answer: The correction is made in introduction section line 35.

L42-44 Suggest deleting sentence beginning "Due to lack". It mostly repeats what's already been said.

Answer: The whole sentence beginning "Due to lack" is deleted from the introduction as suggested.

L1 add "the" before addition

Answer: The correction is made in introduction section line 45.

Section 2.1

L11 Lighting not lightening

Answer: "Lighting" from "lightening" is corrected in section 2.1., line 10.

Section 3.2.1

Thank you for adding sample sizes to section 3.1.2. Please do the same here. NB that percentages are not sample sizes.

Answer: The sample sizes in section 3.2.1. are added wherever previously not added. Samples sizes were added in line 45, page 3 and line 4, page 4.

L42 Change to, "...only 8.3% survived to the blastocyst..."

Answer: The sentence is corrected from "only 8.3% made it to the blastocyst..." to "only 8.3% survived to the blastocyst..." as suggested in section 3.2.1., line 42.